# Hypertension Prevalence and Determinants among Black South African Adults in Semi-Urban and Rural Areas

**DOI:** 10.3390/ijerph17207463

**Published:** 2020-10-14

**Authors:** Peter M. Mphekgwana, Nancy Malema, Kotsedi D. Monyeki, Tebogo M. Mothiba, Mpsanyana Makgahlela, Nancy Kgatla, Irene Makgato, Tholene Sodi

**Affiliations:** 1Research Administration and Development, University of Limpopo, Polokwane 0700, South Africa; 2Department of Psychology, University of Limpopo, Polokwane 0700, South Africa; nancy.malema@ul.ac.za (N.M.); mpsanyana.makgahlela@ul.ac.za (M.M.); irene.makgato@ul.ac.za (I.M.); tholene.sodi@ul.ac.za (T.S.); 3Department of Physiology and Environmental Health, University of Limpopo, Polokwane 0700, South Africa; kotsedi.monyeki@ul.ac.za; 4Faculty of Health Science, University of Limpopo, Polokwane 0700, South Africa; tebogo.mothiba@ul.ac.za; 5Department of Nursing Science, University of Limpopo, Polokwane 0700, South Africa; nancy.kgatla@gmail.com

**Keywords:** hypertension, non-communicable diseases, cardiovascular diseases, semi-urban, rural

## Abstract

The burden of hypertension is reported to be on the rise in developing countries, such as South Africa, despite increased efforts to address it. Using a cross-sectional study design, we assessed and compared the prevalence of and risk factors associated with hypertension amongst adults aged ≥18 years in semi-urban and rural communities (1187 semi-urban and 1106 rural). Trained community health workers administered the INTERHEART Risk Score tool and performed blood pressure assessments using the MEDIC Pharmacists Choice Blood Pressure Monitor. Hypertension was defined to be a systolic blood pressure (BP) ≥ 140 mmHg and diastolic BP ≥ 90 mmHg. A multivariate logistic regression model was used to identify factors and determine their relationship with hypertension. The prevalence of hypertension amongst semi-urban and rural communities was 21% with no gender difference. In the semi-urban area, physical activity, family history, fruit intake, salty food, and eating meat were significantly associated with the odds of hypertension among women, whereas only the waist-to-hip ratio (WHR), diabetic status, and salty food were the predictors for rural women. Factors such as fried food and low fruit intake were significantly associated with the odds of hypertension among men in the semi-urban area, whereas only the WHR was significant among men in the rural area. Hypertension was found to be prevalent among semi-urban and rural adults in Limpopo Province, South Africa.

## 1. Introduction

The world is facing an epidemic of non-communicable diseases (NCDs), such as cardiovascular diseases (CVDs), chronic obstructive pulmonary diseases (COPDs), and cancers, with disproportionately higher levels in the developing countries [1]. It has been estimated that by 2020, 80% of the disease burden will come from NCDs, which contribute 70% of the deaths in developing countries [2]. The rise in NCDs has been attributed to economic growth in developing countries and shifts in societal norms and behaviors, such as dietary habits and physical activity [2,3]. NCDs, mainly cardiovascular diseases, are now a major public health problem that is associated with negative health outcomes [2,4]. Hypertension and diabetes, which are considered the leading causes of death worldwide, are the main modifiable risk factors for cardiovascular disease, which represents the top cause of death worldwide [5,6,7].

The burden of hypertension is reported to be on the rise in developing countries, such as South Africa, despite increased efforts to address it [8]. Although safe and effective antihypertensive medications have been available for decades, hypertension control rates remain low in South Africa when compared with high-income countries [8,9]. Several studies have reported that hypertension is common in both rural and urban South African populations due to lifestyle changes [10,11,12]. In 2005, the prevalence of hypertension was estimated at 25% [13] in rural areas of the country. This had risen to 38% by 2016 [12]. An unhealthy diet, physical inactivity, and aging populations are thought to be the major drivers of the increasing prevalence of hypertension in South Africa [10,12]. Several population-based studies have also reported a high prevalence of hypertension among rural black South African females compared to males [9,12].

Hypertension prevalence in rural adult populations ranges from 25 to 38% [12,13]. Although few previous studies have investigated hypertension and associated risk factors in some rural areas in South Africa, we are not aware of studies that have sought to determine and compare the prevalence of hypertension among semi-urban and rural South African adults. Therefore, the purpose of this study was (1) to compare and determine the prevalence of hypertension among semi-urban and rural communities, and (2) to investigate the factors associated with hypertension in a semi-urban and rural sample of black South Africans from the age of 18 to 101 years.

## 2. Materials and Methods

### 2.1. Study Design and Setting

A population-based cross-sectional study was conducted between July 2019 and January 2020 in the Polokwane Municipality located within the Capricorn District of Limpopo Province. Polokwane Municipality constitutes 60% (797,127 population) of Capricorn District’s population [14]. We conveniently and purposively selected two areas that fall within Polokwane Municipality, namely, Seshego (classified as semi-urban) and Ga-Molepo (classified as rural). Seshego is divided into eight residential zones and is located 9.7 km from the Polokwane city. In contrast, Ga-Molepo consists of more than 15 villages and is located approximately 53 km from the city of Polokwane.

### 2.2. Study Population and Sampling

The majority of residents (approximately 99%) in these identified areas are Black Africans [14]. Assuming a confidence of 95%, a margin of error of 5%, and a conservative prevalence estimate of 50%, the initial sample size was set at 385 for the study. For each study site, the sample size was adapted using a finite population correction factor (FPC). A total of 3193 participants were randomly selected to take part in the study. A total of 2256 people (semi-urban = 1164, rural = 1092) completed the questionnaire, of which 937 of the returned forms were incomplete. Incomplete questionnaires were removed from the study and all persons ≥18 years of age who lived permanently in these specified areas and chose to participate or fill out the consent form were entitled to participate.

### 2.3. Data Collection

The study used a self-administered INTERHEART Risk Score tool for data collection. INTERHEART is a large, international, standardized, case-control study designed as an initial step to assess the importance of risk factors for coronary heart disease involving 52 different countries [15,16]. Sixty-four trained community health workers used the INTERHEART Risk Score tool to screen the participants. The INTERHEART Risk Score tool asks questions about the following: gender, age, smoking, high blood pressure, diabetes, family history of heart attack, diet, stress, depression, and physical activity. For smoking, the participants were classified as: never smoked; if smoking, number of cigarettes smoked per day; former smoker (last smoked more than 12 months ago); exposure to second-hand smoke in the last 12 months for less or more than an hour per week. For physical activity, the subjects were classified as inactive or performs mild exercise (requiring minimal effort) and moderate or strenuous physical activity in their leisure time. In the process of data collection, the community health workers surveyed the participants at their homes. For depression/stress, the participant was asked whether there was ever a time when they felt sad, blue, or depressed for two weeks or more in a row during the past 12 months.

### 2.4. Measurement

The blood pressure was measured on the left-hand arm using the MEDIC Pharmacists Choice Devices Blood Pressure Monitor Classic (MPCDBPMC, Dis-Chem, Midrand, South Africa). Those whose systolic blood pressure measured ≥140 mmHg and diastolic pressure ≥90 mmHg were regarded as having hypertension [17]. This blood pressure monitor has been clinically validated according to the European Society of Hypertension (ESH) protocol [18]. The waist-to-hip ratio (WHR) was measured using a tape measure.

### 2.5. Statistical Analysis

A multivariate logistic regression model was applied to identify factors and determine their relationships with the hypertension variable. A similar methodology has been used elsewhere [11]. In the model, hypertension was the target variable and the risk factors were the explanatory variables (gender, age category, smoking status, WHR, physical activity, depression/stress, and diet). SPSS version 26.0 (IBM SPSS Statistics, Armonk, NY, USA) was used to perform the descriptive analyses (frequency, percentages, and cross-tabulation) and chi-squared tests were used to compare sets of nominal data that had larger frequency counts, while Fisher’s exact test was used when the frequencies were small (less than five or ten). The STATA software (Stata 9.0, StataCorp, College Station, TX, USA) was then used to run a multivariate logistic regression model.

### 2.6. Ethical Considerations

Ethical clearance was obtained from the University of Limpopo Turlfloop Research and Ethics Committee (TREC/381/2017) on 6 February 2019 and permission to conduct the study was obtained from the Limpopo Department of Health Research Committee.

### 2.7. Validity and Reliability

Validity was ensured by using a validated non-laboratory INTERHEART Risk Score tool [15,16]. The INTERHEART Risk Score tool was translated by an expert in translation into Northern Sotho, which is an indigenous language spoken by the majority of the participants. To ensure reliability, the community health workers were trained by professional nurses for a week and for a day before the pilot study on the prevention of CVD and how to screen communities using the INTERHEART Risk Score tool, respectively. Training continued at different clinics after the pilot using the translated INTERHEART Risk Score tool.

## 3. Results

In the total population, the prevalence of waist-to-hip ratios greater than or equal to 0.964 was significantly higher in participants residing in the rural areas (*p* < 0.001) than in participants residing in the semi-urban areas. The prevalence of depressed/stressed participants, parents with a history of heart attack, and low fruit and vegetable consumption (<1 time/day) were significantly higher amongst the rural residents (*p* < 0.001). On the other hand, the prevalences of high consumption of red meat and salty and fried foods three times a day were significantly higher amongst the semi-urban residents (*p* < 0.001). As for age, smoking history, and physical activity, there was no statistically significant difference between the two sites (Table 1).

Table 2 and Table 3 show the general characteristics of the study participants and the prevalence of hypertension amongst residents in semi-urban and rural areas. The prevalence of hypertension amongst the semi-urban and rural communities was 21% with no gender difference. The proportion of hypertension increased with advancing age (more than 35% for participants aged over 55 years). Those who performed mild exercise were found to be more hypertensive than those who performed moderate exercise in the two research sites. Most semi-urban participants whose waist-to-hip ratio ranged from 0.873 to 0.963 and were former smokers were found to be hypertensive. On the other hand, rural participants whose WHR was 0.964 or higher and had never smoked cigarettes tended to be more hypertensive.

The multivariate logistic regression analysis showed a significant association for hypertensive women with a high WHR; diabetes status; physical activity; a history of a parent having a heart attack; consumption of meat, fruits, vegetable, salt, and fried food. After adjusting for age, variables such as WHR, diabetes status, and fried food and vegetable consumption showed an insignificant association with hypertension. Women performing moderate exercise were found to be 0.54 times less likely to be hypertensive than women performing mild exercise. Women who do not consume any fruit daily and whose parents have had a heart attack were three times more likely to be hypertensive than women who consumed one or more fruits daily and their family did not have any heart failure. On the other hand, a history of a parent having a heart attack and the consumption of fruits and fried food showed a significant association with hypertension in men. After adjusting for age, only a history of a parent having a heart attack showed an insignificant association with hypertension (Table 4).

Before and after adjusting for age, the models showed a significant association between hypertension in women with high WHR, diabetes status, and salt intake. Women with high WHR (between 0.873–0.963) were found to be 1.75 times more likely to be hypertensive than women with low WHR (less than 0.873), whereas those who were diabetic were found to be three times more likely to be hypertensive than non-diabetic women. On the other hand, men with a high WHR (greater than 0.964) were found to be four times more likely to be hypertensive (Table 5).

## 4. Discussion

The study aimed to compare and determine the prevalence of hypertension among residents in semi-urban and rural settings, and to investigate its associated risk factors. The overall prevalence of hypertension in the present study was 21% in both areas (semi-urban and rural areas). Previous studies reported a higher level of hypertension in the black female population in South Africa [9,12]. The present findings contradict the previous findings as no significant gender difference was found. Our findings also lend support to previous studies, which indicated that hypertension was prevalent among older rural adults [12,19], with most men (over the age of 55 years) and women (over the age of 65 years) being hypertensive.

The findings further showed that fried food was one of the predictors for hypertension in men residing in a semi-urban area. Contrary to this finding, fried food was found not to be an important predictor for hypertension in rural areas. The difference might be attributable to a possible middle-class status for people in semi-urban areas who may prefer fast food and Western dietary patterns [20]. The Western dietary patterns are characterized by a high intake of salt, red meat, refined sugars, and saturated fat [20,21]. Consistent with previous studies [20,21], this study found that meat and salty food contributed to hypertension among women in the semi-urban area in South Africa.

Moderate exercise, in contrast to mild exercise, was found to be significant at reducing the high risk of being hypertensive among women in the semi-urban area. This finding is consistent with the results of previous studies conducted in South Africa and elsewhere [20,22,23], as they found that a lack of physical activity was significantly associated with hypertension. Some studies have shown that doing 30 min of physical activity five times per week reduces the risk of hypertension [24,25]. However, no significant association was detected between physical activity and hypertension among the rural population. An earlier study reported similar results [26], finding no significant relationship between physical activity and hypertension among the rural population.

A previous study by Lashkardoost et al. found the highest correlation between WHR and changes in hypertension among women in an urban area [27]. Contrary to this finding, in the present study, no association was found between WHR and hypertension among women or men in a semi-urban area. However, the study found a significant association between WHR and hypertension among women (0.873 ≤ WHR ≤ 0.963) and men (WHR ≥ 0.964) in the rural area.

Hypertension and depression share common pathways, with each of these conditions having an impact on the natural history of the other [28]. Our analysis has shown that depression/stress was not an important risk factor for hypertension among the semi-urban and rural populations in South Africa. Similar results were reported in the previous study by Wiehe et al. [29]. This is in contrast with previous studies [8,30], as they suggested that individuals experiencing depression are at high risk of developing hypertension. The discrepancies between the studies’ results might be because the present study used self-reported cases of depression/stress, which has some limitations since some might not be aware that they are experiencing depression/stress.

The worldwide low intake of fruits and vegetables causes an increase in ischemic heart disease, strokes, and cardiovascular diseases [31,32,33]. High consumption of fruit and vegetables reduces the risk of heart disease, blood pressure, and some forms of tumor [21]. In agreement, this study found that low fruit intake was significantly associated with the odds of hypertension among the semi-urban population. Contrary to this finding, among the rural population, no significant association was observed. Similar results were reported in previous rural population studies [12,34].

Earlier studies found that smokers and drinkers have a higher prevalence rate of hypertension than non-smokers and non-drinkers [19]. Our study showed no significant relationship between smoking and hypertension among black semi-urban and rural populations. This is in agreement with previous studies [12,34], which have all found that there is no association between smoking and hypertension.

The findings revealed that among women in rural communities, there is a significant relationship between diabetic status and hypertension, with the odds of hypertension being higher with diabetic participants. This is in agreement with the previous study, as they reported that the odds of hypertension are high with a longer duration of type 2 diabetes mellitus (T2DM) [35]. Women in the semi-urban area with a family history of heart failure were at high risk of hypertension.

Despite several strengths of the study, including a large sample, limitations should be noted. The survey was cross-sectional and was not conducted throughout the year. This means that no causal relationship can be inferred between any of the factors and hypertension. Self-reported answers may be exaggerated and respondents may be too embarrassed to reveal private details or have forgotten some of their statuses, which can contribute to bias in the results. Finally, the fact that more women than men engaged in the study acts as a drawback of the study. As a recommendation, we suggest that future studies should include a fair representation of both genders.

## 5. Conclusions

In this study, we established the prevalence of hypertension among semi-urban and rural adults in the Polokwane Municipality, Limpopo Province, South Africa, with different contributing risk factors. Based on these findings, it is therefore suggested that screening and monitoring for blood pressure among the semi-urban and rural population should be encouraged. Furthermore, there is a need to promote hypertension awareness and interventions, such as regular exercise to reduce potential cardiovascular complications.

## Figures and Tables

**Table 1 ijerph-17-07463-t001:** Prevalence of demographic characteristics and behavioral risk factors for hypertension by residential area.

Variables	Semi-Urban (*n* = 1164)	Rural (*n* = 1092)	*p*-Value
	*n* (%)	*n* (%)	
Gender			<0.001
Female	864 (74)	832 (76)	
Male	300 (26)	260 (24)	
Age (years)			0.422
18–25	146 (13)	132 (12)	
26–35	162 (14)	182 (17)	
36–55	381 (33)	381 (35)	
>55	478 (41)	384 (36)	
Smoking history			0.113
Non-smoker	1027 (90)	932 (88)	
Former smoker	57 (5)	45 (4)	
Current smoker	61 (5)	84 (8)	
Waist-to-hip ratio (WHR)			<0.001
<0.873	760 (66)	612 (56)	
0.873–0.963	316 (27)	333 (31)	
≥0.964	84 (7)	140 (13)	
Physical activity			0.151
Mild exercise	744 (64)	724 (66)	
Moderate exercise	416 (36)	368 (34)	
Depressed/stressed	481 (41)	336 (31)	<0.001
Parent had a heart attack	62 (5)	87 (8)	<0.001
High salty food consumption (≥1 time/day)	968 (83)	782 (72)	<0.001
High fried food/trans saturated fat consumption (≥3 times/week)	944 (81)	584 (53)	<0.001
Low fruit consumption (<1 time/day)	145 (12)	359 (33)	<0.001
Low vegetable consumption (<1 time/day)	155 (13)	247 (23)	<0.001
Red meat/poultry consumption (≥2 times/day)	860 (74)	703 (64)	<0.001

**Table 2 ijerph-17-07463-t002:** Prevalence of hypertension of semi-urban residents aged ≥18.

Semi-Urban Area	Normotensive	Hypertensive (*n* = 254, 21%)	Proportion with Hypertension	*p*-Value
Gender				0.072
Female	665	199	23.03%	
Male	248	52	17.33%	
Age (years)				<0.001
18–25	139	7	5%	
26–35	153	9	6%	
36–55	320	61	16%	
>55	306	172	36%	
Smoking history				0.928
Non-smoker	806	221	21.52%	
Former smoker	44	13	22.81%	
Current smoker	49	12	19.67%	
WHR				<0.001
<0.873	635	125	16.45%	
0.873–0.963	216	100	31.65%	
≥0.964	61	23	27.38%	
Physical activity				0.002
Mild exercise	562	182	24.46%	
Moderate exercise	346	70	16.83%	
Depressed/stressed	409	72	14.97%	<0.001
Parent had a heart attack	38	24	38.71%	0.001
High salty food consumption (≥1 time/day)	799	169	17.46%	<0.001
High fried food/trans saturated fat consumption (≥3 times/week)	785	159	16.84%	<0.001
Low fruit consumption (<1 time/day)	126	19	13.10%	0.007
Low vegetable consumption (<1 time/day)	129	26	16.77%	0.225
Red meat/poultry consumption (≥2 times/day)	713	147	17.09%	<0.001

**Table 3 ijerph-17-07463-t003:** Prevalence of hypertension of rural residents aged ≥18.

Rural Area	Normotensive	Hypertensive	Proportion with Hypertension	*p*-Value
		(*n* = 235, 21%)		
Gender				0.602
Female	650	182	21.88%	
Male	209	51	19.62%	
Age				<0.001
18–25	125	7	5%	
26–35	169	13	7%	
36–55	328	53	14%	
>55	224	160	42%	
Smoking history				0.267
Non-smoker	727	205	22.00%	
Former smoker	40	5	11.11%	
Current smoker	66	18	21.43%	
WHR				<0.001
<0.873	515	97	15.85%	
0.873–0.963	243	90	27.03%	
≥0.964	95	45	32.14%	
Physical activity				0.136
Mild exercise	562	162	22.38%	
Moderate exercise	300	68	18.48%	
Depressed/stressed	257	79	23.51%	0.004
Parent had a heart attack	70	17	19.54%	
High salty food consumption (≥1 time/day)	631	151	19.31%	<0.001
High fried food/trans saturated fat consumption (≥3 times/week)	473	111	19.01%	0.126
Low fruit consumption (<1 time/day)	284	75	20.89%	0.970
Low vegetable consumption (<1 time/day)	201	46	18.62%	0.437
Red meat/poultry consumption (≥2 times/day)	547	156	22.19%	0.583

**Table 4 ijerph-17-07463-t004:** Multivariate logistic regression model to determine the predictors of hypertension in the semi-urban area.

Variable	Response	Unadjusted		Adjusted for Age	
		Women	Men	Women	Men
		Odds (95% CI)	Odds (95% CI)	Odds (95% CI)	Odds (95% CI)
Smoking history	Non-smoker	7.807 (0.1335, 4.5645)	1.5444 (0.5637, 4.2310)	1.0795 (0.1743, 6.6845)	1.3524 (0.4627, 3.9533)
	Former smoker	0.9426 (0.1028, 8.6394)	1.2722 (0.3332, 4.8570)	1.5003 (0.1365, 16.4873)	1.4372 (0.3620, 5.7058)
	Current smoker				
WHR	<0.873				
	0.873–0.963	1.6448 (1.0595, 2.5533) *	1.6203 (0.6927, 3.7897)	1.5298 (0.9580, 2.4429)	1.5243 (0.6118, 3.7976)
	≥0.964	2.7541 (1.2828, 5.9128) *	1.0800 (0.3029, 3.8495)	1.5935 (0.7006, 3.6244)	1.0684 (0.2764, 4.1302)
Diabetes	No				
	Yes	2.6595 (1.3323, 5.3088) *	3.4157 (0.6991, 16.6883)	1.9339 (0.9422, 3.9693)	2.6098 (0.5078, 13.4108)
Physical activity	Mild exercise				
	Moderate exercise	0.5538 (0.3531, 0.8684) *	0.6106 (0.2560, 1.4564)	0.5438 (0.3387, 0.8732) *	0.5827 (0.2358, 1.4399)
Depressed/stressed	No				
	Yes	0.7659 (0.4972, 1.1797)	0.5087 (0.2115, 1.2237)	0.7133 (0.4497, 1.1316)	0.5120 (0.2064, 1.2703)
Parent had a heart attack	No				
	Yes	2.9675 (1.3944, 6.3152) *	4.3190 (1.0082, 8.5009) *	3.6247 (1.6388, 8.0170) *	4.2559 (0.9149, 19.7980)
High salty food consumption (≥1 time/day)	Yes				
	No	0.2986 (0.1831, 0.4869) *	0.7645 (0.2899, 2.0155)	0.3417 (0.2014, 0.5799) *	0.9761 (0.3434, 2.7738)
High fried food/trans saturated fat consumption (≥3 times/week)	Yes				
	No	0.5138 (0.3139, 0.8412) *	0.2638 (0.1005, 0.6925) *	0.6252 (0.3644, 1.0727)	0.2004 (0.0699, 0.5745) *
Low fruit consumption (<1 time/day)	No				
	Yes	2.6858 (1.1684, 6.1738) *	9.3997 (1.5636, 12.5065) *	2.9861 (1.2297, 7.2507) *	12.1032 (1.6091, 19.033) *
Low vegetable consumption (<1 time/day)	No				
	Yes	2.3573 (1.0278, 5.4064) *	0.5223 (0.1487, 1.8336)	1.3659 (0.5544, 3.3651)	0.3937 (0.1019, 1.5215)
Red meat/poultry consumption (≥2 times/day)	No				
	Yes	0.4634 (0.2969, 0.7231) *	0.7025 (0.2947, 1.6744)	0.5194 (0.3209, 0.8405) *	0.7677 (0.2999, 1.9649)

* Statistically significant (*p*-value < 0.05).

**Table 5 ijerph-17-07463-t005:** Multivariate logistic regression model to determine the predictors of hypertension in the rural area.

Variable	Response	Unadjusted		Adjusted for Age	
		Women	Men	Women	Men
		Coefficient (95% CI)	Coefficient (95% CI)	Coefficient (95% CI)	Coefficient (95% CI)
Smoking history	Non-smoker	1.5178 (0.4751, 4.8489)	0.5052 (0.1811, 1.4087)	2.0751 (0.6295, 6.8406)	0.6081 (0.1912, 1.9342)
	Former smoker	0.2827 (0.0205, 3.8850)	0.2949 (0.0578, 1.5048)	0.3131 (0.0230, 4.2483)	0.3663 (0.0617, 2.1716)
	Current smoker				
WHR	<0.873				
	0.873–0.963	2.1963 (1.4083, 3.4252) *	1.9184 (0.6941, 5.3021)	1.7480 (1.0824, 2.8230) *	1.2098 (0.3845, 3.8060)
	≥0.964	1.7549 (0.9078, 3.3923)	7.0548 (2.4288, 20.4910) *	1.4243 (0.7059, 2.8738)	4.6198 (1.3152, 16.2275) *
Diabetes	No				
	Yes	5.5058 (2.7372, 11.0748) *	3.3133 (0.9674, 11.3479)	3.4884 (1.6796, 7.2451) *	3.6728 (0.8830, 15.2762)
Physical activity	Mild exercise				
	Moderate exercise	0.9412 (0.5968, 1.4843)	0.7907 (0.3379, 1.8504)	0.8591 (0.5313, 1.3891)	−0.5432 (−1.5550, 0.4195)
Depressed/stressed	No				
	Yes	1.1370 (0.7406, 1.7457)	0.7201 (0.2774, 1.8690)	1.0262 (0.6523, 1.6146)	0.5116 (0.1712, 1.5287)
Parent had a heart attack	No				
	Yes	0.4103 (0.1512, 1.1136)	0.3060 (0.0592, 1.5810)	0.5709 (0.1992, 1.6358)	0.3711 (0.0585, 2.3548)
High salty food consumption (≥1 time/day)	Yes				
	No	0.4549 (0.2871, 0.7208) *	0.6057 (0.2284, 1.6059)	0.4856 (0.2979, 0.7913) *	0.6571 (0.2244, 1.9238)
High fried food/trans saturated fat consumption (≥3 times/week)	Yes				
	No	1.0462 (0.6636, 1.6496)	0.9343 (0.3736, 2.3366)	1.2883 (0.7892, 2.1030)	0.9302 (0.3212, 2.6935)
Low fruit consumption (<1 time/day)	No				
	Yes	1.1233 (0.6920, 1.8233)	0.7012 (0.2586, 1.9014)	1.0628 (0.6354, 1.7776)	0.7203(0.2306, 2.2502)
Low vegetable consumption (<1 time/day)	No				
	Yes	1.1535 (0.6586, 2.0200)	1.0489 (0.3442, 3.1962)	0.9148 (0.5001, 1.6732)	0.6423 (0.1805, 2.2854)
Red meat/poultry consumption (≥2 times/day)	No				
	Yes	1.4587 (0.9062, 2.3479)	0.6660 (0.2517, 1.7620)	1.3968 (0.8448, 2.3093)	0.7842 (0.2553, 2.4090)

* Statistically significant (*p*-value < 0.05).

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
