# Peer review of "Hypertension Prevalence and Determinants among Black South African Adults in Semi-Urban and Rural Areas"

_ijerph, 2020, doi:10.3390/ijerph17207463_

Round 1

Reviewer 1 Report

The authors improved n the areas that I had suggested in my original review.

Author Response

Author's appreciates the positive comment.

Reviewer 2 Report

Authors have addressed almost all points raised by reviewers. Please add the depression/stress criteria to the text:

"For depression/stress, the participant was asked if during the past 12 months, was there ever a time
when they felt sad, blue, or depressed for two weeks or more in a row"

Author Response

The author's added depression/stress criteria in the text.

Reviewer 3 Report

The paper improved significantly. 

I have just some remarkrs.

  1. Please, specify that you used multivariate logistic regression, no need to say it was binary.
  2. Please, presenr at the beginning of the results section infornation on the overall prevalence in the studied population. You did this in an Abstract, but not in the body of the paper.
  3. It's still not mentioned why did you include much less men than women. It has to be mentioned as limitation of the study

Author Response

1) The author's changed binary to multivariate in the text.

2) The author's added the overall prevalence in the studied population in the body.

3) The author's added this in the limitation.

This manuscript is a resubmission of an earlier submission. The following is a list of the peer review reports and author responses from that submission.

Round 1

Reviewer 1 Report

Materials and Methods Section 2.2: The sample size calculation only required 133 participants per area yet the sample is 10 times that size. Why? How were the participants recruited? Were there any inclusion or exclusion criteria? What was the response rate?

2.3 Provide reference for INTERHEART risk score  studies so reader understands how the measurements were collected. Fr example was BP measured once, twice or three times and then the average used. What is the validity and reliability of this risk score? Add that to this section as well as references. Where was the data collected?

2.4  Add the p-value level of significance. How did you decide what variables were entered into the CHAID? Was this based on the chi-sq cross tabulations presented in tables 1 and 2? Explain rationale for the WHR cut points, usuallay .85 fo rfemales and .90 for males. What is the rationale fo rthe age cut points. Explain rationale.

2.6

Mentions a pilot study- what is this? Is this manuscript based on a pilot study?

Results:

Because the purpose of the study was to compare HTN in two communities, the presentation of the tables can be improved. First, a table 1 should report the characteristics of the study population, overall, not stratified by HTN status. This will help the reader to understand the population characteristics.

Table  Characteristics of the sample

Characteristics  Seshego Ga-Molepo -p-value  
  n(%) n(%)    
Age (yrs)        

Table 2  Characteristics of the population stratified by HTN status in Seshego

Characteristic Overall

Normotensive

n(%)

HTN

N(%)

p-value
Age (yrs)        

Table 3 same as table 2 but for Ga-Molepo

If space is an issue, then combine tables 2 and 3 into 1 table.

4 Discussion

Line 188- not sure how you drew this conclusion based on the data.

Line 226 indicates that ou had a high response rate, but it is not reported in text. What was the response rate and how was it calculated?

Line 229 indicates bias may have been present. Explain how this might have affected the results.

Reviewer 2 Report

Mphekgwana et al. reported the prevalence of hypertension and its determinants among black south African adults in semi-urban and rural areas. The study has some interesting data but there are several points which need to be addressed in the current form. Major points: 1) Majority of participants were females in both areas. This can significantly affect the contribution of other risk factors. Sub analysis of the risk factors effects with regard to hypertension need to be conducted in male and female populations. 2) Why age groups are different in male and females (i.e. 55 years vs 65 years)? 3) Regular exercise has been recommended by authors to reduce hypertension among adults by authors whereas their data does not support that (no sig effect of exercise was observed in the study for Ga-Molepo area) 4) INTERHEART Risk Score tool needs to be presented in the paper. what is considered mild and moderate exercise? How depression or being stressed is assessed? IS it based on medication etc? Minor points: Font is not consistent. Please check that. Lines 169-172 need revision.

Reviewer 3 Report

I'd like to congratulate study team for their efforts in conducting this large and interesting study.

Abstract

Lines 28-29. Please, see remark below on non-significance of difference in prevalence among men and women.

I recommend to skip the last sentence of the abstract as your study did not address such issue as public awareness of the disease.

Study population

It’s better to use “18 and more” or like that description here and below in the results unless you decided to include only up to 101 old and not older.

It is unclear how exactly you enrolled individuals. What was the way to invite them to participate – phoning, some advertising? Did individuals come to any medical setting or were they’ve been visited by study staff?

Line 93.  The INTERHEART Risk Score is mentioned in this line, but references are given. I would suggest to use references in the line 93.

Results section

Please, report the overall prevalence of disease among the whole population (men and women), then discuss gender related data.

Please, for both tables 1 and 2 as well as for text better use not the settlements names but indicate which population lives there – semi-urban or rural.

The title of columns 2 and 3 in both tables should be presented in similar way – normotensive/hypertensive or no hypertension/hypertension.

Line 133. % is missed after 17.33

Lines 131-133. This is not right to say that the majority of hypertensive patients were female. At first, there were much bigger number of female recruited that male. Secondly, the difference is not really significant. The prevalence is nearly the same among men and women and this is confirmed by statistical tool also. No significance found according to table 1. The comment in the text should be that no gender difference was found.

Lines 143 and 148. What are those values related to?

Line 149. Please, describe here, that Fig. 1 is about semi-urban and Fig. 2 about rural people

Lines 151-155. You should describe here to what population (i.e. semi-urban) theses data are related.

Line 153. Which are those five important variables. It seems something missed in the next statement.

Line 161. No need to repeat what are presented on the Fig. 1. You did it before.

Discussion section

Lines 183-185. As I mentioned before, this is not correct to say that women are more affected. You didn’t confirm this statistically. It seems, the prevalence is near the same. So, your data are not like previously published.

Why so few men were enrolled, comparing with women? This has to be explained and discussed also. And this has to be mentioned as an example of possible source of bias when you present study limitations.

Reviewer 4 Report

The topic about hypertension prevalence and determinants has been widely studied worldwide, even in in different cities in South Africa. Thus, it is a big challenge for the research novelty.

For hypertension determinants, the current decision tree model is not ideal one. Some analysis methods, such as Logistic regression and dominance analysis, may be more effective in discovering the differences in factors between the semi-urban and rural areas.